# MobileViTv3: Mobile-Friendly Vision Transformer with Simple and Effective Fusion of Local, Global and Input Features

## Abstract

MobileViT (MobileViTv1) combines convolutional neural networks (CNNs) and vision transformers (ViTs) to create light-weight models for mobile vision tasks. Though the main MobileViTv1-block helps to achieve competitive state-of-the-art results, the fusion block inside MobileViTv1-block creates scaling challenges and has a complex learning task. We propose changes to the fusion block that are *simple and effective* to create MobileViTv3-block, which addresses the scaling and simplifies the learning task. Our proposed MobileViTv3-block used to create MobileViTv3-XXS, XS and S models outperform MobileViTv1 on ImageNet-1k, ADE20K, COCO and PascalVOC2012 datasets. On ImageNet-1K, MobileViTv3-XXS and MobileViTv3-XS surpasses MobileViTv1-XXS and MobileViTv1-XS by 2% and 1.9% respectively. Recently published MobileViTv2 architecture removes fusion block and uses linear complexity transformers to perform better than MobileViTv1. We add our proposed fusion block to MobileViTv2 to create MobileViTv3-0.5, 0.75 and 1.0 models. MobileViTv3-0.5 and MobileViTv3-0.75 outperforms MobileViTv2-0.5 and MobileViTv2-0.75 by 2.1% and 1.0% respectively on ImageNet-1K dataset. For segmentation task, MobileViTv3-1.0 achieves 2.07% and 1.1% better mIOU compared to MobileViTv2-1.0 on ADE20K dataset and PascalVOC2012 dataset respectively. Our code and the trained models will be made available on GitHub.

## 1 Introduction

Convolutional Neural Networks (CNNs) [ResNet (He et al., 2016), DenseNet (Huang et al., 2017) and EfficientNet (Tan & Le, 2019)] are widely used for vision tasks such as classification, detection and segmentation, due to their strong performance on the established benchmark datasets such as Imagenet (Russakovsky et al., 2015), COCO (Lin et al., 2014), PascalVOC (Everingham et al., 2015), ADE20K (Zhou et al., 2017) and other similar datasets. When deploying CNNs on edge devices like mobile phones which are generally resource constrained, light-weight CNNs suitable for such environments come from family of models of MobileNets (MobileNetv1, MobileNetv2, MobileNetv3) (Howard et al., 2019), ShuffleNets (ShuffleNetv1 and ShuffleNetv2) (Ma et al., 2018) and light-weight versions of EfficientNet (Tan & Le, 2019) (EfficientNet-B0 and EfficientNet-B1). These relatively small models lack in accuracy when compared to models with large parameters and FLOPs. Recently, Vision Transformers (ViTs) have emerged as an strong alternatives to CNNs on these vision tasks. Self-attention mechanism in ViTs interacts with all parts of the image to produce features which have global information embedded in them. This has been demonstrated to produce comparable results to CNNs but with large pre-training data and advance data augmentation (Dosovitskiy et al., 2020). Also, this global processing comes at a cost of large parameters and FLOPs to match the performance of CNNs as seen in ViT (Dosovitskiy et al., 2020), and its different versions such as DeiT (Touvron et al., 2021), SwinT (Liu et al., 2021), MViT (Fan et al., 2021), Focal-ViT (Yang et al., 2021), PVT (Wang et al., 2021), T2T-ViT (Yuan et al., 2021b), XCiT (Ali et al., 2021).

Many recent work have introduced convolutional layers in ViT architecture to form hybrid networks to improve performance, achieve sample efficiency and make the models more efficient in terms of parameters and FLOPs like MobileViTs (MobileViTv1 (Mehta & Rastegari, 2021), MobileViTv2

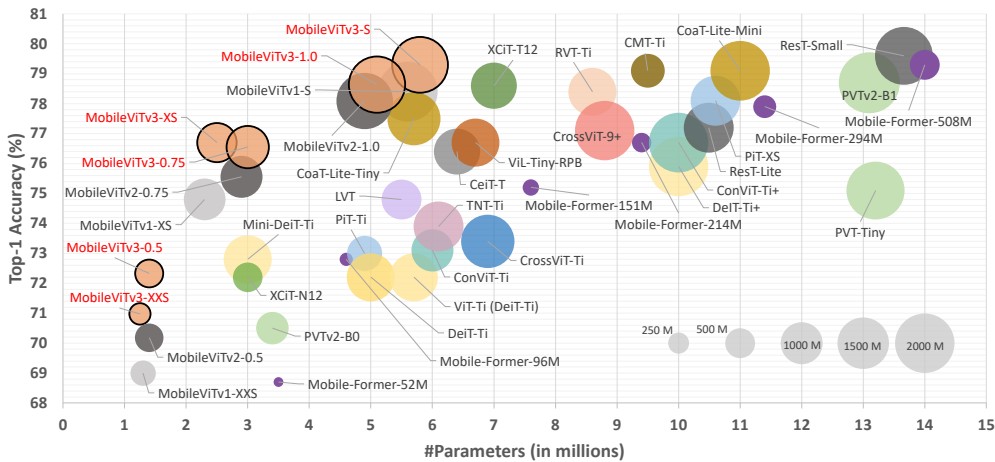

Figure 1: Comparing Top-1 accuracies of MobileViTv3, ViT variants and hybrid models on ImageNet-1K dataset. The area of bubbles correspond to number of FLOPs in the model. The reference FLOP sizes are shown in the bottom right (example, 250M is 250 Mega-FLOPs/Million-FLOPs). Models of our MobileViTv3 architecture outperforms other models with similar parameter budget of under 2M, 2-4M and 4-8M. Also, they achieve competitive results when compared to the models greater than 8M parameters.

Mehta & Rastegari (2022) ), CMT (Guo et al., 2022), CvT (Wu et al., 2021), PVTv2 (Wang et al., 2022), ResT (Zhang & Yang, 2021), MobileFormer (Chen et al., 2022), CPVT (Chu et al., 2021), MiniViT (Zhang et al., 2022), CoAtNet (Dai et al., 2021), CoaT (Xu et al., 2021a). Performance of many of these models on ImageNet-1K, with parameters and FLOPs is shown in Figure 1. Among these models, only MobileViTs and MobileFormer are specifically designed for resource constrained environment such as mobile devices. These two models achieve competitive performance compared to other hybrid networks with less parameters and FLOPs. Even though these small hybrid models are critical for the vision tasks on mobile devices, there is little work done in this area.

Our work focuses on improving one such light-weight family of models known as MobileViTs (MobileViTv1 (Mehta & Rastegari, 2021) and MobileViTv2 (Mehta & Rastegari, 2022)). When compared to the models with parameter budget of 6 million(M) or less, MobileViTs achieve competitive state-of-the-art results with a simple training recipe (basic data augmentation) on classification task. Also it can be used as an efficient backbone across different vision tasks such as detection and segmentation. While focusing on only the models with 6M parameters or less, we pose the question: Is it possible to change the model architecture to improve its performance by maintaining similar parameters and FLOPs? *To do so, our work looks into challenges of MobileViT-block architecture and proposes **simple and effective** way to fuse input, local (CNN) and global (ViT) features which lead to significant performance improvements on Imagenet-1K, ADE20k, PascalVOC and COCO dataset.* We propose four main changes to MobileViTv1 block (three changes w.r.t MobileViTv2 block) as shown in figure 2. Three changes are in the fusion block: First, 3x3 convolutional layer is replaced with 1x1 convolutional layer. Second, features of local and global representation blocks are fused together instead of input and global representation blocks. Third, input features are added in the fusion block as a final step before generating the output of MobileViT block. Fourth change is proposed in local representation block where normal 3x3 convolutional layer is replaced by depthwise 3x3 convolutional layer. These changes result in the reduction of parameters and FLOPs of MobileViTv1 block and allow scaling (increasing width of the model) to create a new MobileViTv3-S, XS and XXS architecture, which outperforms MobileViTv1 on classification (Figure 1), segmentation and detection tasks. For example, MobileViTv3-XXS and MobileViTv3-XS perform 2% and 1.9% better with similar parameters and FLOPs on ImageNet-1K dataset compared to MobileViTv1-XXS and MobileViTv1-XS respectively. In MobileViTv2, fusion block is absent. Our proposed fusion block is introduced in MobileViTv2 architecture to create MobileViTv3-1.0, 0.75 and 0.5 architectures. MobileViTv3-0.5 and MobileViTv3-0.75 outperforms MobileViTv2-0.5 and MobileViTv2-0.75 by 2.1% and 1.0% respectively with similar parameters and FLOPs on ImageNet-1K dataset.

## 2 RELATED WORK

**Vision Transformers**: ViT (Dosovitskiy et al., 2020) introduced the transformer models used for Natural Language Processing tasks to vision domain, specifically for image recognition. Later, its different versions such as DeiT (Touvron et al., 2021) improved the performance by introducing a novel training technique and reducing the dependency on large pre-training data. Works focusing on improving self-attention mechanism to boost performance include XCiT (Ali et al., 2021), SwinT (Liu et al., 2021), ViL (Zhang et al., 2021) and Focal-transformer (Yang et al., 2021). XCiT introduces cross-covariance attention where self-attention is operated on feature channels instead of tokens. SwinT modified the ViT to make it a general purpose architecture for classification, detection and segmentation tasks. ViL improves ViT by encoding image at multiple scales and uses self-attention mechanism which is a variant of Longformer (Beltagy et al., 2020). Recent works like T2T-ViT (Yuan et al., 2021b) and PVT (PVTv1) (Wang et al., 2021) also focus on introducing CNN like hierarchical feature learning by reducing spatial resolution or token sizes of output after each layer. Few new architectures like CrossViT (Chen et al., 2021), MViT (Fan et al., 2021), MViTv2 (Li et al., 2022) and Focal-transformer (Yang et al., 2021) learn both local features (features learnt specifically from neighbouring pixels/features/patches) and global features (features learnt using all pixels/features/patches).

**CNNs**: Models like EfficientNet-B7 (Tan & Le, 2019), ConvNeXt (Liu et al., 2022), EfficientNetV2 (Tan & Le, 2021), RegNetY (Radosavovic et al., 2020) and NFNet-F4+ (Brock et al., 2021) achieve high accuracy on ImageNet-1K dataset and also many can be used as a general purpose backbone models for detection and segmentation tasks. But, these CNN models are generally high in number of parameters and FLOPs. Light-weight CNNs that achieve competitive performance with less parameters and FLOPs include EfficientNet-B0 and B1, MobileNetV3 (Howard et al., 2019), ShuffleNetv2 (Ma et al., 2018) and ESPNetv2 (Mehta et al., 2019). EfficientNet studied model scaling and developed family of efficientnet models which are still one of the most efficient CNNs in terms of parameters and FLOPs. MobileNetV3 belongs to category of models specifically developed for resource constrained environments such as Mobile phones. Building block of MobileNetV3 architecture uses MobileNetv2 (Sandler et al., 2018) block and Squeeze-and-Excite (Hu et al., 2018) network in it. ShuffleNetv2 studies and proposes guidelines for efficient model design and produces shufflenetv2 family of models which also performs competitively with other light-weight CNN models. ESPNetv2 uses depth-wise dilated separable convolution to create EESP (Extremely Efficient Spatial Pyramid) unit which helps to reduce parameters and FLOPs and achieve competitive results.

**Hybrids**: Lately, several models are being proposed to combine CNNs and ViTs together in one architecture to capture both long-range dependencies using self-attention mechanism of ViT and local information using local kernels in CNNs to improve performance on vision tasks. MobileViT (MobileViTv1 (Mehta & Rastegari, 2021), MobileViTv2 Mehta & Rastegari (2022)) and Mobile-Former (Chen et al., 2022) have been specifically designed for constrained environments like mobile devices. CMT (Guo et al., 2022) architecture has convolutional stem and stacks convolutional layers and transformer layers alternatively. CvT (Wu et al., 2021) uses convolutional token embedding instead of linear embedding and a convolutional transformer layer block that leverages these convolutional token embeddings to improve performance. CoAtNet (Dai et al., 2021) unifies depthwise convolution and self-attention using simple relative attention and vertically stacks convolutional layers and attention layers. CeiT (Yuan et al., 2021a) introduces locally-enhanced-feed-forward layer by using depth-wise convolution with other changes to achieve competitive results. ViTAE (Xu et al., 2021b) has convolution layers in parallel to multi-head self-attention module and both are fused and fed to feedforward network, also ViTAE uses convolutional layers to embed inputs to token. RVT (Mao et al., 2022) uses convolutional stem to generate patch embeddings and uses convolutional feed-forward network in transformer to achieve better results.

## 3 NEW MOBILEVIT ARCHITECTURE

Our work proposes four design changes to the existing MobileViTv1 block architecture to build MobileViTv3-block as shown in Figure 2a. Section 3.1 explains these four changes in MobileViTv3-block architecture and compares with MobileViTv1 and MobileViTv2-blocks. Section 3.2 details MobileViTv3-S, XS and XXS architectures and shows how it is scaled compared to MobileViTv1-S, XS and XXS. In recently published MobileViTv2 architecture, changes applied to MobileViT-

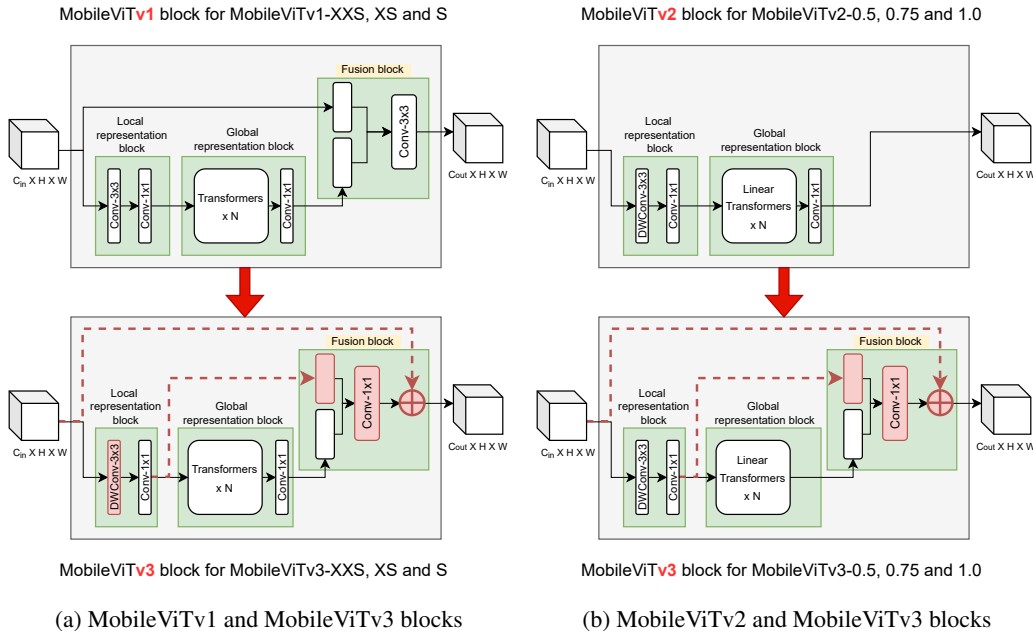

Figure 2: A comparison between: (a) MobileViTv1 and MobileViTv3 modules, and (b) Mobile-ViTv2 and MobileViTv3 modules. The proposed architectural changes are highlighted in red.

block are, fusion block is removed, transformer block uses self-attention with linear complexity and depthwise convolutional layer is used in local representation block. We add back the fusion block to MobileViTv2-block with our proposed changes to create MobileViTv3-block as shown in Figure 2b for MobileViTv3-0.5, 0.75, 1.0 architectures.

## 3.1 MOBILEVITV3 BLOCK

**Replacing 3x3 convolutional layer with 1x1 convolutional layer in fusion block**: Two main motivations exist for replacing 3x3 convolutional layer in fusion. *First, fuse local and global features independent of other locations in the feature map to simplify the fusion block's learning task.* Conceptually, 3x3 convolutional layer is fusing input features, global features, and other location's input and global features which are present in the receptive field, which is a complex task. Fusion block's goal can be simplified by allowing it to fuse input and global features, independent of other locations in feature map. To do so, we use 1x1 convolutional layer in fusion instead of 3x3 convolutional layer. *Second, is to remove one of the major constraints in scaling of MobileViTv1 architecture.* Scaling MobileViTv1 from XXS to S is done by changing width of the network and keeping depth constant. Changing width (number of input and output channels) of MobileViTv1 block causes large increase in number of parameters and FLOPs. For example, if the input and output channels are doubled (2x) in MobileViTv1 block, the number of input channels to 3x3 convolutional layer inside fusion block increases by 4x and output channels by 2x, because input to the 3x3 convolutional layer is concatenation of input and global representation block features. This causes a large increase in parameters and FLOPs of MobileViTv1 block. Using 1x1 convolutional layer avoids this large increase in parameters and FLOPs while scaling. **Local and Global features fusion**: In fusion layer, features from *local* and global representation blocks are concatenated in our proposed MobileViTv3 block instead of *input* and global representation features. This is because the local representation features are more closely related to the global representation features when compared to the input features. The output channels of the local representation block are slightly higher than the channels in input features. This causes an increase in the number of input feature maps to the fusion block's 1x1 convolutional layer, but the total number of parameters and FLOPs are significantly less than the baseline MobileViTv1 block due to the change of 3x3 convolutional layer to 1x1 convolutional layer. **Fusing input features**: Input features are added to the output of 1x1 convolutional layer in the fusion block. The residual connections in models like ResNet and DenseNet have shown to help

the optimization of deeper layers in the architecture. By adding the input features to the output in the fusion block, we introduce this residual connection in new MobileViTv3 architecture. Ablation study results shown in table 5 demonstrates that this residual connection contributes 0.6% accuracy gain. **Depthwise convolutional layer in local representation block**: To further reduce parameters, 3x3 convolutional layer in local representation block is replaced with depthwise 3x3 convolutional layer. As seen in the ablation study results table 5, this change does not have a large impact on the Top-1 ImageNet-1K accuracy gain and provides good parameter and accuracy trade-off.

## 3.2 SCALING UP BUILDING BLOCKS

Applying the changes proposed in section 3.1, allows scaling of our MobileViTv3 architecture by increasing the width (number of channels) of the layers. Table 1 shows MobileViTv3-S, XS and XXS architectures with their output channels in each layer, scaling factor, parameters an FLOPs.

Table 1: MobileViTv3-S, XS and XXS architecture details and comparison with MobileViTv1-S, XS and XXS. Values given in brackets '()' represent scaling factor compared to MobileViTv1 models.

| Layer | Size | Stride | Repeat | XXS | XS | S |
|---|---|---|---|---|---|---|
| Image | 256x256 | 1 | | | | |
| Conv-3x3, $\downarrow$ 2 | 128x128 | 2 | 1 | 16 | 16 | 16 |
| MV2 | 128x128 | 2 | 1 | 16 | 32 | 32 |
| MV2 , $\downarrow$ 2 | 64x64 | 4 | 1 | 24 | 48 | 64 |
| MV2 | 64x64 | 4 | 2 | 24 | 48 | 64 |
| MV2 , $\downarrow$ 2 | 32x32 | 8 | 1 | 64 (1.3x) | 96 (1.5x) | 128 (1.3x) |
| MobileViT block (L=2) | 32x32 | 8 | 1 | 64 (1.3x) | 96 (1.5x) | 128 (1.3x) |
| MV2 , $\downarrow$ 2 | 16x16 | 16 | 1 | 80 (1.3x) | 160 (2.0x) | 256 (2.0x) |
| MobileViT block (L=4) | 16x16 | 16 | 1 | 80 (1.3x) | 160 (2.0x) | 256 (2.0x) |
| MV2 , $\downarrow$ 2 | 8x8 | 32 | 1 | 128 (1.6x) | 160 (1.7x) | 320 (2.0x) |
| MobileViT block (L=3) | 8x8 | 32 | 1 | 128 (1.6x) | 160 (1.7x) | 320 (2.0x) |
| Conv-1x1, | 8x8 | 32 | 1 | 512 (1.6x) | 640 (1.7x) | 1280 (2.0x) |
| Global pool | 1x1 | 256 | 1 | 512 | 640 | 1280 |
| Linear | 1x1 | 256 | 1 | 1000 | 1000 | 1000 |
| Parameters (M) | | | | 1.25 | 2.5 | 5.8 |
| FLOPs (M) | | | | 289 | 927 | 1841 |

## 4 EXPERIMENTAL RESULTS

Our work shows results on classification task using ImageNet-1K in section 4.1, segmentation task using ADE20K and PASCAL VOC 2012 datasets in section 4.2, detection task using COCO dataset in section 4.3. We also discuss changes to our proposed MobileViTv3 architecture for improving latency and throughput in appendix B.

## 4.1 IMAGE CLASSIFICATION ON IMAGENET-1K

**Implementation details**: Except for the batch size, hyperparameters used for MobileViTv3-S, XS and XXS are similar to the MobileViTv1 and hyperparameters used for MobileViTv3-1.0, 0.75 and 0.5 are similar to MobileViTv2. Due to resource constraints, we were limited to using a total batch size of 384 (32 images per GPU) for experiments on MobileViTv3-S and XS. To maintain consistency in batch sizes, MobileViTv3-XXS is also trained on batch size of 384. Batch size of 1020 (85 images per GPU) used of MobileViTv3-0.5,0.75 and 1.0 training. More hyperparameter details in appendix C.1. Performance is evaluated using single crop top-1 accuracy, for inference an exponential moving average of model weights is used. All the classification models are trained from scratch on the ImageNet-1K classification dataset. This dataset contains 1.28M and 50K images for training and validation respectively.

**Comparison with MobileViTs**: Table 2 demonstrates that performance of all the versions of Mo-bileViTv3 surpass MobileViTv1 and MobileViTv2 versions with similar parameters and FLOPs and smaller training batch size. Increasing total batch size from 192 to 384 improves accuracy of

MobileViTv3-S, XS and XXS models (Table 2). This indicates the potential for further accuracy gains with batch size of 1024 on MobileViTv3-XXS, XS and S models. It is also important to note that MobileViTv3-S, XS and XXS models trained with basic data augmentation not only outperforms MobileViTv1-S, XS, XXS, but also surpasses performance of MobileViTv2-1.0, 0.75 and 0.5 which are trained with advanced data augmentation.

Table 2: MobileViT V1, V2 and V3 comparison in terms of Top-1 ImageNet-1k accuracy, parameters and operations. Models with similar parameters and operations are grouped together for clear comparison.

| Model | Training Batch size | FLOPs (M)↓ | # Params. (M)↓ | Top-1 (%)↑ |
|---|---|---|---|---|
| MobileViTv1-XXS | 1024 | 364 | 1.3 | 69.00 |
| MobileViTv3-XXS | 192 | 289 | 1.2 | 70.02 (+1%) |
| **MobileViTv3-XXS** | 384 | 289 | 1.2 | **70.98** (+2%) |
| MobileViTv2-0.5 | 1024 | 466 | 1.4 | 70.18 |
| **MobileViTv3-0.5** | 1020 | 481 | 1.4 | **72.33** (+2.1%) |
| MobileViTv1-XS | 1024 | 986 | 2.3 | 74.8 |
| MobileViTv3-XS | 192 | 927 | 2.5 | 76.3 (+1.5%) |
| **MobileViTv3-XS** | 384 | 927 | 2.5 | **76.7** (+1.9%) |
| MobileViTv2-0.75 | 1024 | 1030 | 2.9 | 75.56 |
| **MobileViTv3-0.75** | 1020 | 1064 | 3.0 | **76.55** (+0.99%) |
| MobileViTv1-S | 1024 | 2009 | 5.6 | 78.4 |
| MobileViTv3-S | 192 | 1841 | 5.8 | 78.8 (+0.4%) |
| **MobileViTv3-S** | 384 | 1841 | 5.8 | **79.3** (+0.9%) |
| MobileViTv2-1.0 | 1024 | 1851 | 4.9 | 78.09 |
| **MobileViTv3-1.0** | 1020 | 1876 | 5.1 | **78.64** (+0.55%) |

**Comparison with ViTs**: Figure 1 compares our proposed MobileViTv3 models performance with other ViT variants and hybrid models. Following MobileViTv1, we mainly compare our models with parameter budget of around 6M or less. Also, when comparing to models greater than 6M parameters, we limit FLOPs budget to ∼2 GFLOPs or less because our largest model in this work has ∼2 GFLOPs. *Models under 2 million parameters*: To the best of our knowledge, only MobileViT variants exist in this range. MobileViTv3-XXS and MobileViTv3-0.5 outperform other MobileViT variants. MobileViTv3-0.5 by far achieves the best accuracy of 72.33 %. *Models between 2-4 million parameters*: MobileViTv3-XS and MobileViTv3-0.75 outperform all the models in this range. Top-1 accuracy of MobileViTv3-XS on ImageNet-1k is 76.7%, which is 3.9% higher than Mini-DeiT-Ti (Zhang et al., 2022) and 4.5 % higher than XCiT-N12 (Ali et al., 2021). Although Mobile-Former-53M (Chen et al., 2022) uses only 53 GFLOPs, it lags in accuracy by a large margin of 12.7% with MobileViTv3-XS. *Models between 4-8 million parameters*: MobileViTv3-S attains the highest accuracy of 79.3% in this parameter range. MobileViTv3-S with simple training recipe and 300 training epochs is 0.7% better than XCiT-T12 trained using distillation, advanced data augmentation and 400 epochs. It is 1.8% and 2.6% better than Coat-Lite-Tiny (Xu et al., 2021a) and ViL-Tiny-RPB (Zhang et al., 2021) respectively. MobileViTv3-S is 1% better with 0.5x FLOPs and similar parameters as compared to CoaT-Tiny (Xu et al., 2021a). *Models greater than 8 million parameters*: We also compare our designed models with existing models having more than 8M parameters and around 2 GFLOPs. When compared with MobileViTv3-S (79.3%) trained with basic data augmentation and 300 epochs, MobileFormer-508M achieves similar accuracy of 79.3% with ∼2.5x more parameters, ∼3.5x less FLOPs, advance data augmentation and 450 training epochs, CMT-Ti (Guo et al., 2022) achieves 79.1% with ∼1.6x more parameters, ∼2.9x less FLOPs (due to input image size of 160x160) and advanced data augmentation.

**Comparison with CNNs**: Figure 3 compares our proposed models with the CNN models which are light-weight with a parameter budget of ∼6M or less, similar to MobileViTv1 (Mehta & Rastegari, 2021). *Models in 1-2 million parameters range*: MobileViTv3-0.5 and MobileViTv3-XXS with 72.33% and 70.98% respectively are best accuracies in this range. MobileViTv3-0.5 achieves over 2.5% improvement compared to ESPNetv2-123M, MobileNetv3-small(0.75) (Howard et al., 2019) and MobileNetV2(0.5) (Sandler et al., 2018). *Models with 2-4 million parameters*: MobileViTv3-XS achieves over 4% improvement compared to MobileNetv3-Large(0.75), ShuffleNetv2(1.5), ESPNetv2-284M and MobileNetv2(0.75). *Models with 4-8 million parameters*: MobileViTv3-S shows more than 2% accuracy gain over EfficientNet-B0 (Tan & Le, 2019), ESPNetv2-602M

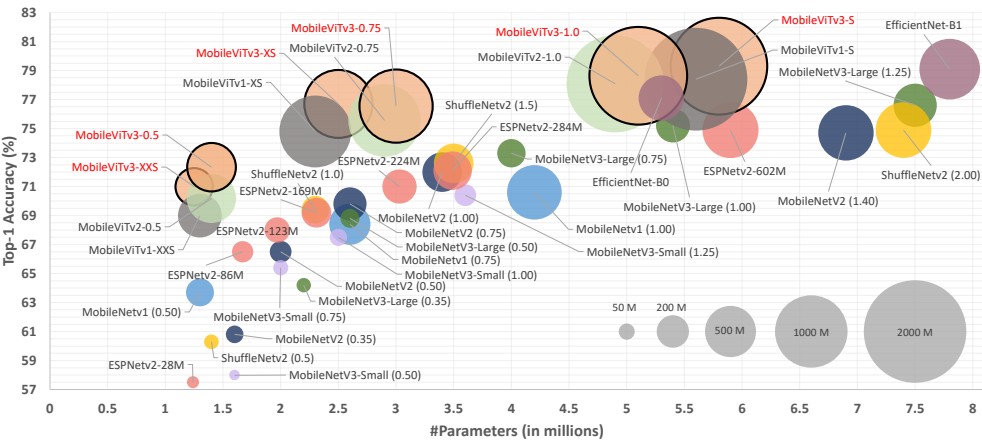

Figure 3: Top-1 accuracy comparison between MobileViTv3 models and existing light-weight CNN models on ImageNet-1K dataset. The bubble size corresponds to the number of FLOPs. The reference FLOP sizes are shown in the bottom right (example, 50M is 50 Mega-FLOPs/Million-FLOPs). Models of our MobileViTv3 architecture outperform other models with similar parameter budget of 1-2M, 2-4M and 4-8M.

and MobileNetv3-Large(1.25). EfficientNet-B1 with 1.3x more parameters and 2.6x less FLOPs achieves competitive accuracy of 79.1% compared to MobileViTv3-S with accuracy of 79.3%.

## 4.2 SEGMENTATION

**Implementation details**: The segmentation performance is evaluated on the validation set and reported using mean Intersection over Union (mIOU). ***PASCAL VOC 2012 dataset***: Following MobileViTv1, MobileViTv3 is integrated with DeepLabv3 (Chen et al., 2017) for segmentation task on PASCAL VOC 2012 dataset (Everingham et al., 2015). Extra annotations and data is used from (Hariharan et al., 2011) and (Lin et al., 2014) respectively, which is a standard practise for training on PascalVOC2012 dataset (Chen et al., 2017); (Mehta et al., 2019). For MobileViTv3-S, XS and XXS, training hyperparameters are similar to MobileViTv1 except the batch size. Smaller batch size of 48 (12 images per GPU) is used compared to 128 (32 images per GPU) for MobileViTv1. For MobileViTv3-1.0, 0.75 and 0.5, all the hyperparameters are kept same as used for MobileViTv2-1.0, 0.75 and 0.5 training. ***ADE20K dataset*** (**Zhou et al., 2019**): Contains total 25K images with 150 semantic categories. Out of 25K images, 20K images are used for training, 3K images for test and 2K images for validation. Same training hyperparameters are used for MobileViTv2 models as MobileViTv3-1.0, 0.75 and 0.5 models.

**Results**: ***PASCAL VOC 2012 dataset***: Table 3a demonstrates MobileViTv3 models with lower training batch size of 48, outperforming their corresponding counterpart models of MobileViTv1 and MobileViTv2 which are trained on higher batch size of 128. MobileViTv3-1.0 achieves 80.04% mIOU, which outperforms MobileViTv2-1.0 by 1.1%. MobileViTv3-XS is 1.6% better than MobileViTv1-XS and MobileViTv3-0.5 surpasses MobileViTv2-0.5 by 1.41%. ***ADE20K dataset***: Table 3b shows the results of MobileViTv3-1.0, 0.75 and 0.5 models on ADE20K dataset. MobileViTv3-1.0, 0.75 and 0.5 models outperform MobileViTv2-1.0, 0.75 and 0.5 models by 2.07%, 1.73% and 1.64% respectively.

## 4.3 OBJECT DETECTION

**Implementation details**: MS-COCO dataset (Lin et al., 2014) with 117K training and 5K validation images, is used to evaluate the detection performance of MobileViTv3 models. Similar to MobileViTv1, we integrated pretrained MobileViTv3 as a backbone network in Single Shot Detection network (SSD) (Liu et al., 2016) and the standard convolutions in the SSD head are replaced with separable convolutions to create SSDLite network. This SSDLite with pre-trained MobileViTv3

Table 3: Comparing MobileViTv3 segmentation task results on PASCAL VOC 2012 and ADE20K datasets. # params of MobileViT models denotes the number of parameters of the encoder/backbone architecture only.

(a) Segmentation on PASCAL VOC 2012 dataset

| Backbone | # Params (M)↓ | mIOU(%) ↑ |
|---|---|---|
| MobileViTv1-XXS | 1.9 | 73.6 |
| **MobileViTv3-XXS** | 1.96 | **74.04** (+0.44%) |
| MobileViTv2-0.5 | 6.2 | 75.07 |
| **MobileViTv3-0.5** | 6.3 | **76.48** (+1.41%) |
| MobileViTv1-XS | 2.9 | 77.1 |
| **MobileViTv3-XS** | 3.3 | **78.77** (+1.6%) |
| MobileViTv1-S | 6.4 | 79.1 |
| **MobileViTv3-S** | 7.2 | **79.59** (+0.49%) |
| MobileViTv2-1.0 | 13.32 | 78.94 |
| **MobileViTv3-1.0** | 13.56 | **80.04** (+1.10%) |

(b) Segmentation on ADE20K dataset

| Backbone | # Params (M)↓ | mIOU(%) ↑ |
|---|---|---|
| MobileViTv2-0.5 | 6.31 | 31.93 |
| **MobileViTv3-0.5** | 6.37 | **33.57** (+1.64%) |
| MobileViTv2-0.75 | 9.6 | 34.7 |
| **MobileViTv3-0.75** | 9.71 | **36.43** (+1.73%) |
| MobileViTv2-1.0 | 13.4 | 37.06 |
| **MobileViTv3-1.0** | 13.62 | **39.13** (+2.07%) |

backbone is fine-tuned on MS-COCO dataset. No change to training hyperparameters. Detailed list of hyperparameters in appendix C.2. Performance evaluation is done on validation set using mAP@IoU of 0.50:0.05:0.95 metric.

**Results**: Table 4a and 4b show the detection results on COCO dataset. # params of MobileViT models indicates number of parameters of the encoder/backbone architecture only. MobileViTv3 comparison with other light-weight CNN models is shown in Table 4a. MobileViTv3-XS outperforms MobileViTv1-XS by 0.8% and MNASNet by 2.6% mAP. Comparison with heavy-weight CNNs detailed in Table 4b. MobileViTv3-XS and MobileViTv3-1.0 surpasses MobileViTv1-XS and MobileViTv2-1.0 by 0.8% and 0.5% mAP respectively.

Table 4: Comparing MobileViTv3 detection task results on COCO dataset with light-weight and heavy-weight CNNs. # params of MobileViT models denotes the number of parameters of the encoder/backbone architecture only.

(a) Comparison w/light-weight CNNs

| Backbone | # Params (M)↓ | mAP(%) ↑ |
|---|---|---|
| MobileViTv1-XXS | 1.5 | 18.5 |
| **MobileViTv3-XXS** | 1.53 | **19.3** (↑0.8%) |
| MobileViTv2-0.5 | 2 | 21.24 |
| **MobileViTv3-0.5** | 2 | **21.8** (↑0.56%) |
| MobileViTv2-0.75 | 3.6 | 24.57 |
| **MobileViTv3-0.75** | 3.7 | **25.0** (↑0.43%) |
| MobileNetv3 | 4.9 | 22.0 |
| MobileNetv2 | 4.3 | 22.1 |
| MobileNetv1 | 5.1 | 22.2 |
| MixNet | 4.5 | 22.3 |
| MNASNet | 4.9 | 23.0 (↑0.0%) |
| MobileViTv1-XS | 2.7 | 24.8 (↑1.8%) |
| **MobileViTv3-XS** | 2.7 | **25.6** (↑2.6%) |

(b) Comparison w/heavy-weight CNNs

| Backbone | # Params (M)↓ | mAP(%) ↑ |
|---|---|---|
| VGG | 35.6 | 25.1 |
| ResNet50 | 22.9 | 25.2 (↑0.0%) |
| MobileViTv1-XS | 2.7 | 24.8 (↓0.4%) |
| **MobileViTv3-XS** | 2.7 | **25.6** (↑0.4%) |
| MobileViTv2-1.0 | 5.6 | 26.47 (↑1.27%) |
| **MobileViTv3-1.0** | 5.8 | **27.0** (↑1.8%) |
| MobileViTv1-S | 5.7 | 27.7 (↑2.5%) |
| **MobileViTv3-S** | 5.5 | **27.3** (↑2.1%) |

## 4.4 ABLATION STUDY OF OUR PROPOSED MOBILEVITV3 BLOCK

**Implementation details**: We study the effect of the four proposed changes on MobileViTv1-S block by adding changes one by one. The final model with all the four changes is our unscaled version and we name it: MobileViTv3-S(unscaled). To match the number of parameters of MobileViTv1 we increase the width of MobileViTv3-S(unscaled), giving us MobileViTv3-S. In this ablation study we train models for 100 epochs, use batch size of 192 (32 images per GPU) and other hyper-parameters are default as given in section 4.1. In Table 5, 'conv-3x3': 3x3 convolutional layer in fusion block, 'conv-1x1': 1x1 convolutional layer in fusion block, 'Input-Concat': concatenating input features with global representation in the fusion block, 'Local-Concat': concatenating local-representation block output features with global representation in the fusion block, 'Input-Add': adding input

features to the output of fusion block, 'DWConv': using depthwise convolutional layer in the local representation block and 'Top-1': Top-1 accuracy on ImageNet-1K dataset.

Table 5: Ablation study of MobileViTv3 block. 'unscaled' indicates that the number of channels in the architecture are kept same as the baseline MobileViTv1. ✓represents incorporating the change in the block.

| Model | Conv 3x3 | Conv 1x1 | Input Concat | Local Concat | Input Add | DW Conv | Top-1 (%)↑ |
|---|---|---|---|---|---|---|---|
| MobileViTv1-S | ✓ | | ✓ | | | | 73.7 (↑0.0%) |
| MobileViTv3-S(unscaled) | | ✓ | ✓ | | | | 74.8 (↑1.1%) |
| MobileViTv3-S(unscaled) | | ✓ | | ✓ | | | 74.7 (↑1.0%) |
| MobileViTv3-S(unscaled) | | ✓ | | ✓ | ✓ | | 75.3 (↑1.6%) |
| MobileViTv3-S(unscaled) | | ✓ | | ✓ | ✓ | ✓ | 75.0 (↑1.3%) |

**With 100 training epochs**: Results are shown in Table 5. The baseline MobileViTv1-S, in fusion block, concatenates input features with global representation block features and uses 3x3 convolutional layer. Also, it uses normal 3x3 convolutional layer in the local representation block. This baseline achieves an accuracy of 73.7%. 1. *Replacing 3x3 convolution with 1x1 convolutional layer in fusion block*, MobileViTv3-S(unscaled) achieves 1.1% improvement. This result supports the assumption that simplifying the fusion block's task (allowing fusion layer to fuse local and global features independent of the other location's local and global features) should help optimization to attain better performance. 2. Along with 1x1 convolutional layer in fusion block, *concatenating local representation features instead of input features* results in similar performance gains of 1% compared to concatenating input features. 3. This allows us to incorporate the next change i.e, *to add input features to the output of fusion block* to create a residual connection for helping optimization of deeper layers in the model. With this change, MobileViTv3-S(unscaled) attains 1.6% accuracy gain over the baseline MobileViTv1-S and 0.6% gain over the last change demonstrating the clear advantage of this residual connection. 4. To further reduce number of parameters and FLOPs in MobileViTv3-block, *depth-wise convolutional layer is used instead of normal convolutional layer in the local representation block*. MobileViTv3-S(unscaled) maintains high accuracy gains by achieving 1.3% gain over the baseline. 0.3% accuracy drop can be observed when compared to the previous change. We adopt this change since it reduces parameters and FLOPs without significantly impacting performance and helps in scaling of MobileViTv3-block.

Table 6: MobileViTv3-S(unscaled), MobileViTv1-S and MobileViTv3-S Top-1 ImageNet-1K accuracy comparisons. With similar parameters and FLOPs after scaling, MobileViTv3-S is able to exhibit better performance than baseline MobileViTv1-S.

| Model | Training batch size | FLOPs (M)↓ | # Params (M) ↓ | Top-1 (%)↑ |
|---|---|---|---|---|
| MobileViTv1-S | 192 | 2009 | 5.6 | 75.6 |
| MobileViTv3-S(unscaled) | 192 | 1636 (↓18.6%) | 4.3 (↓22.7%) | 77.5 (↑1.9%) |
| MobileViTv1-S | 1024 | 2009 | 5.6 | 78.4 |
| MobileViTv3-S | 384 | 1841 (↓8.3%) | 5.8 (↑3.6%) | 79.3 (↑0.9%) |

**With 300 epochs training**: Results shown in Table 6. When trained for 300 epochs with the batch size of 192, the baseline MobileViTv1-S achieves Top-1 accuracy of 75.6%, which is lower by 2.8% compared to reported accuracy on MobileViTv1-S trained on 1024 batch size. With all the four proposed changes implemented in MobileViTv1-S architecture to form MobileViTv3-S(unscaled), the model reaches Top-1 accuracy of 77.5%, which outperforms the baseline by 1.9% with 22.7% and 18.6% less parameters FLOPs respectively. MobileViTv3-S(unscaled) architecture though better than the baseline MobileViTv1-S with training batch size of 192, performs worse than the MobileViTv1-S trained at batch size of 1024. Therefore, MobileViTv3-S, XS and XXS models are scaled to have similar parameters and FLOPs as MobileViTv1-S, XS and XXS and are trained with batch size of 384. Table 6 demonstrates that after scaling, MobileViTv3-S is able to outperform MobileViTv1-S by achieving 79.3% accuracy with similar parameters and FLOPs.

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

## A    FINETUNING

Table 7 compares finetuning results of MobileViTv3-1.0, 0.75 and 0.5 with MobileViTv2-1.0, 0.75 and 0.5. Implementation details are similar to section 4.1.

Table 7: Fine-tuned MobileViT V2 and V3 comparison in terms of Top-1 ImageNet-1k accuracy, parameters and operations. <Model name>(384) indicate that the model was fine-tuned using an image size of 384x384. Models with similar parameters and operations are grouped together for clear comparison.

| Model | Training Batch size | FLOPs (M)↓ | # Params. (M)↓ | Top-1 (%)↑ |
|---|---|---|---|---|
| MobileViTv2-0.5 (384) | 64 | 1048 | 1.4 | 72.14 |
| **MobileViTv3-0.5** (384) | 64 | 1083 | 1.4 | **74.01** (+1.87%) |
| MobileViTv2-0.75 (384) | 64 | 2318 | 2.9 | 76.98 |
| **MobileViTv3-0.75** (384) | 64 | 2395 | 3.0 | **77.81** (+0.83%) |
| MobileViTv2-1.0 (384) | 64 | 4083 | 4.9 | 79.68 |
| **MobileViTv3-1.0** (384) | 64 | 4220 | 5.1 | **79.74** (+0.06%) |

# B  IMPROVING LATENCY AND THROUGHPUT

**Implementation details**: We use GeForce RTX 2080 Ti GPU for obtaining latency timings. Results are averaged over 10000 iterations. The timing results may vary $\pm 0.1$ ms. Throughput for XXS, XS and S are calculated on 1000 iterations with batch size of 100. 'Blocks' in Table 8 represents number of MobileViTv3-blocks in 'layer4' of MobileViTv3 architectures (Table 1). *To improve the latency, we reduce the number of MobileViT-blocks in 'layer4' from 4 to 2.*

**Results**: Table 8 shows the latency and throughput results. MobileViTv3-XXS with similar parameters and FLOPs as the baseline MobileViTv1-XXS, along with 1.98% accuracy improvement achieves similar latency of $\sim$7.1 ms. MobileViTv3-XXS with two MobileViT-blocks instead of four, has 30% less FLOPs and achieves latency of 6.24 ms which is $\sim$1 ms faster than the baseline MobileViTv1-XXS. With similar changes in MobileViTv3-XS and MobileViTv3-S architecture, FLOPs are reduced by 13.5% and 17.82% respectively and latency is reduced by $\sim$1 ms and $\sim$0.7 ms respectively.

| Model | Blocks (↓) | FLOPs (↓) | # Params (↓) | Top-1 (↑) | Throughput (↑) | # Time (ms) (↓) |
|---|---|---|---|---|---|---|
| MobileViTv1-XXS | 4 | 364 | 1.3 | 69 | 2124 | 7.24 |
| MobileViTv3-XXS | 4 | 289 | 1.25 | 70.98 (↑1.98%) | 2146 | 7.12 |
| **MobileViTv3-XXS** | **2** | 256 (↓30%) | 1.14 | **70.23** (↑1.23%) | **2308** | **6.24** |
| MobileViTv1-XS | 4 | 986 | 2.3 | 74.8 | 1097 | 7.32 |
| MobileViTv3-XS | 4 | 927 | 2.5 | 76.7 (↑1.9%) | 1078 | 7.2 |
| **MobileViTv3-XS** | **2** | 853 (↓13.5%) | 2.3 | **76.1** (↑1.3%) | **1129** | **6.35** |
| MobileViTv1-S | 4 | 2009 | 5.6 | 78.4 | 822 | 7.34 |
| MobileViTv3-S | 4 | 1841 | 5.8 | 79.3 (↑0.9%) | 824 | 7.29 |
| **MobileViTv3-S** | **2** | 1651 (↓17.82%) | 5.2 | **79.06** (↑0.6%) | **876** | **6.6** |

Table 8: Latency and throughput comparison between MobileViTv3-XXS, XS, and S and MobileViTv1-XXS, XS, and S. While keeping the parameters and Top-1 accuracy similar to MobileViTv1, MobileViTv3 with 2 blocks reduces the number of FLOPs and improves the throughput and latency.

# C  HYPERPARAMETERS

## C.1  CLASSIFICATION

**MobileViTv3-S, XS and XXS**: Default hyperparameters used from MobileViTv1 include using AdamW as optimizer, multi-scale sampler (S = (160,160), (192,192), (256,256), (288,288), (320,320)), learning rate increased from 0.0002 to 0.002 for the first 3K iterations and then annealed to 0.0002 using cosine schedule, L2 weight decay of 0.01, basic data augmentation i.e, random resized cropping and horizontal flipping.

**MobileViTv3-1.0, 0.75 and 0.5**: Default hyperparameters used from MobileViTv2 include using AdamW as optimizer, batch-sampler (S = (256,256)), learning rate increased from 1e-6 to 0.002 for the first 20K iterations and then annealed to 0.0002 using cosine schedule, L2 weight decay of 0.05,

advanced data augmentation i.e, random resized cropping, horizontal flipping, random augmentation, random erase, mixup and cutmix.

## C.2 COCO

**MobileViTv3-S, XS and XXS**: Default hyperparameters include using images of input resolution of 320 x 320, AdamW optimizer, weight decay of 0.01, cosine learning rate scheduler, total batch size of 128 (32 images per GPU), smooth L1 and cross-entropy losses are used for object localization and classification respectively.

**MobileViTv3-1.0, 0.75 and 0.5**: Default hyperparameters include using images of input size 320 x 320, AdamW optimizer, weight decay of 0.05, cosine learning rate scheduler, total batch size of 128 (32 images per GPU), smooth L1 and cross-entropy losses are used for object localization and classification respectively.

## D DISCUSSION AND LIMITATIONS

This work is an effort towards improving performance of models for resource constrained environments like mobile phones. We looked at reducing memory (parameters), computation (FLOPs), latency while boosting accuracy and throughput. With the proposed changes to MobileViT blocks we achieve higher accuracy, with same memory and computation as the baseline MobileViTv1 and v2 as seen in section 4.1. Table 7 shows fine-tuning results which also outperform the fine-tuned MobileViTv2 models. Section B shows how we can achieve better latency and throughput with minimal impact on the accuracy of the model. While MobileViTv3 has higher accuracy and lower or similar parameters as compared to other mobile-CNNs, it's higher FLOPs can be an issue for edge devices (Figure 3). This limitation of MobileViTv3 architecture is inherited from the self-attention module of ViTs. To solve this issue, we will further explore optimization of the self-attention block. Table 2 shows results on Imagenet-1K. The reported accuracies of MobileViTv3-XXS, XS and S models on Imagenet-1K can potentially be further improved by increasing the training batch size to 1024 similar to the baseline model. The proposed fusion of input features, local features (CNN features) and global features (ViT features) shown in this paper can also be explored in other hybrid architectures.

## E OBJECT DETECTION AND SEMANTIC SEGMENTATION RESULTS

### E.1 OBJECT DETECTION ON COCO DATASET

Figure 4 shows object detection results on COCO validation images using SSD-Lite with MobileViTv3-S as its backbone. Figure 5 shows object detection results on COCO validation images using SSD-lite with MobileViTv3-1.0 as its backbone. The images shown in figure 4 include challenging object detection examples (blurred human/person and complex background).

### E.2 SEMANTIC SEGMENTATION ON PASCALVOC2012 DATASET

Figure 6 shows segmentation results on PascalVOC2012 validation images using Deeplabv3 with MobileViTv3-S as its backbone. Figure 7 shows segmentation results on PascalVOC2012 validation images using Deeplabv3 with MobileViTv3-1.0 as its backbone. In figure 6 and 7, moving from left to right we provide the input image, the corresponding segmentation output, and the overlay of segmentation output on the input image.

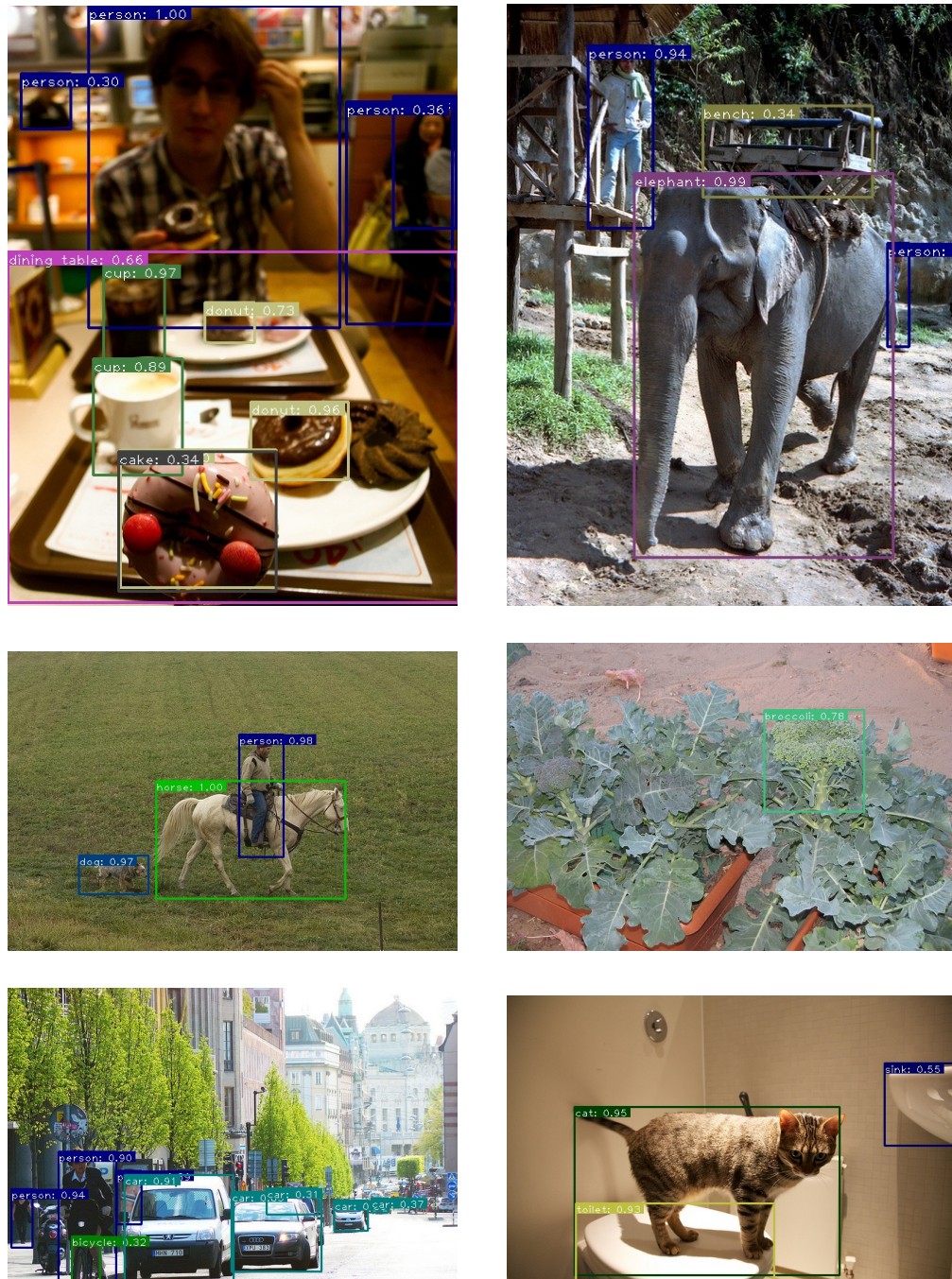

Figure 4: Object detection results using SSD-Lite model with MobileViTv3-S as its backbone.

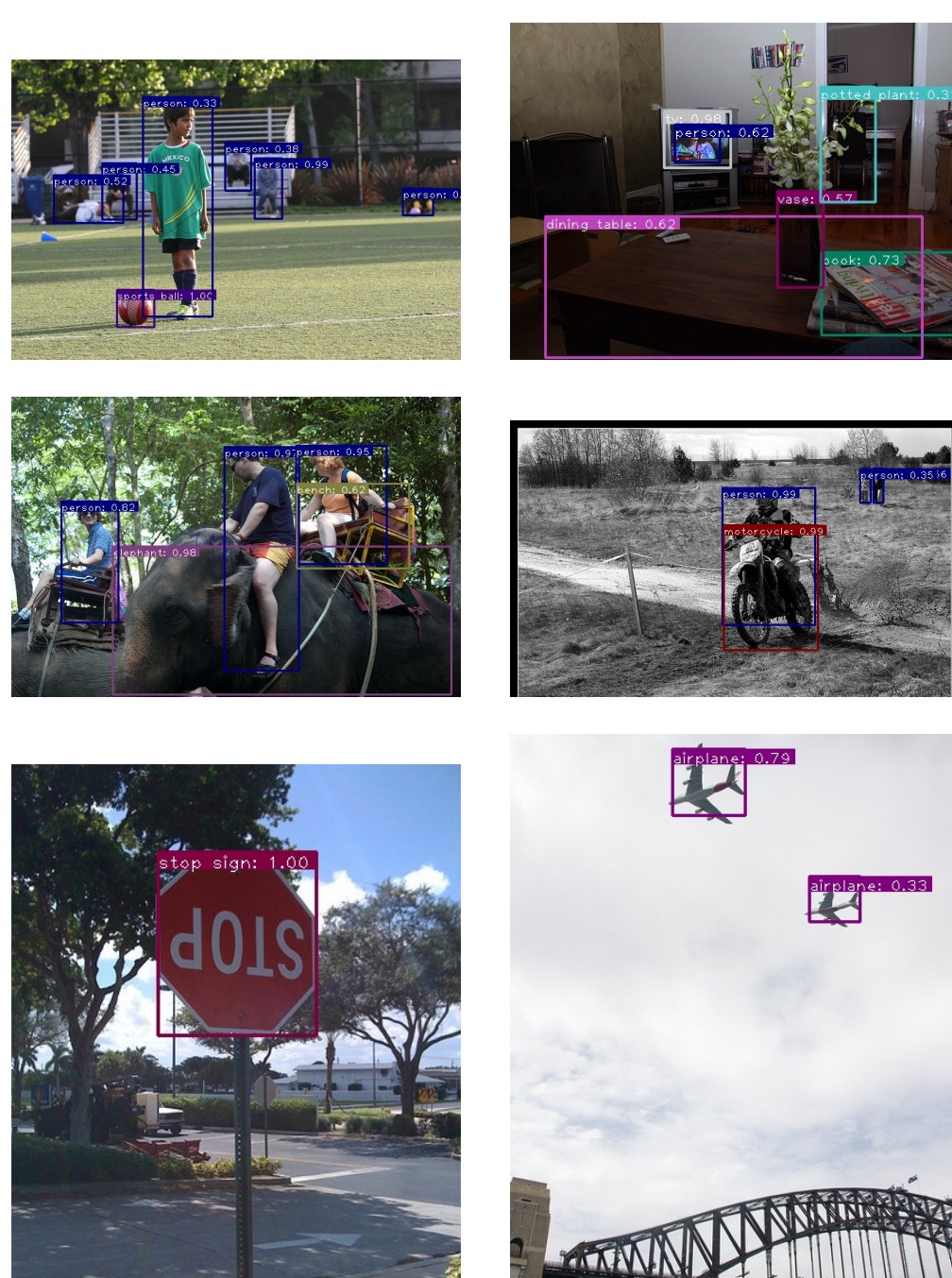

Figure 5: Object detection results using SSD-Lite model with MobileViTv3-1.0 as its backbone.

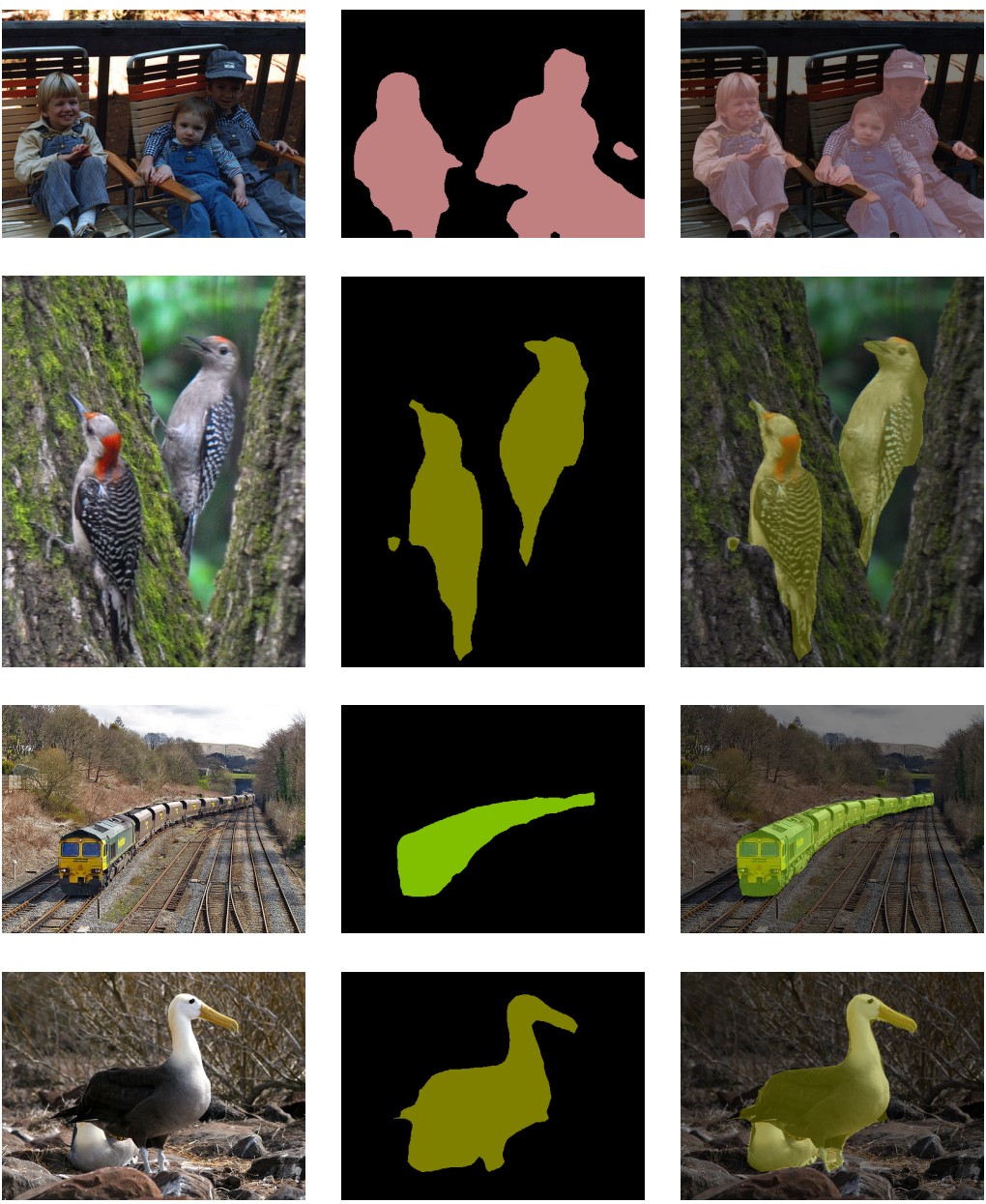

Figure 6: Semantic segmentation results using Deeplabv3 with MobileViTv3-S as its backbone.

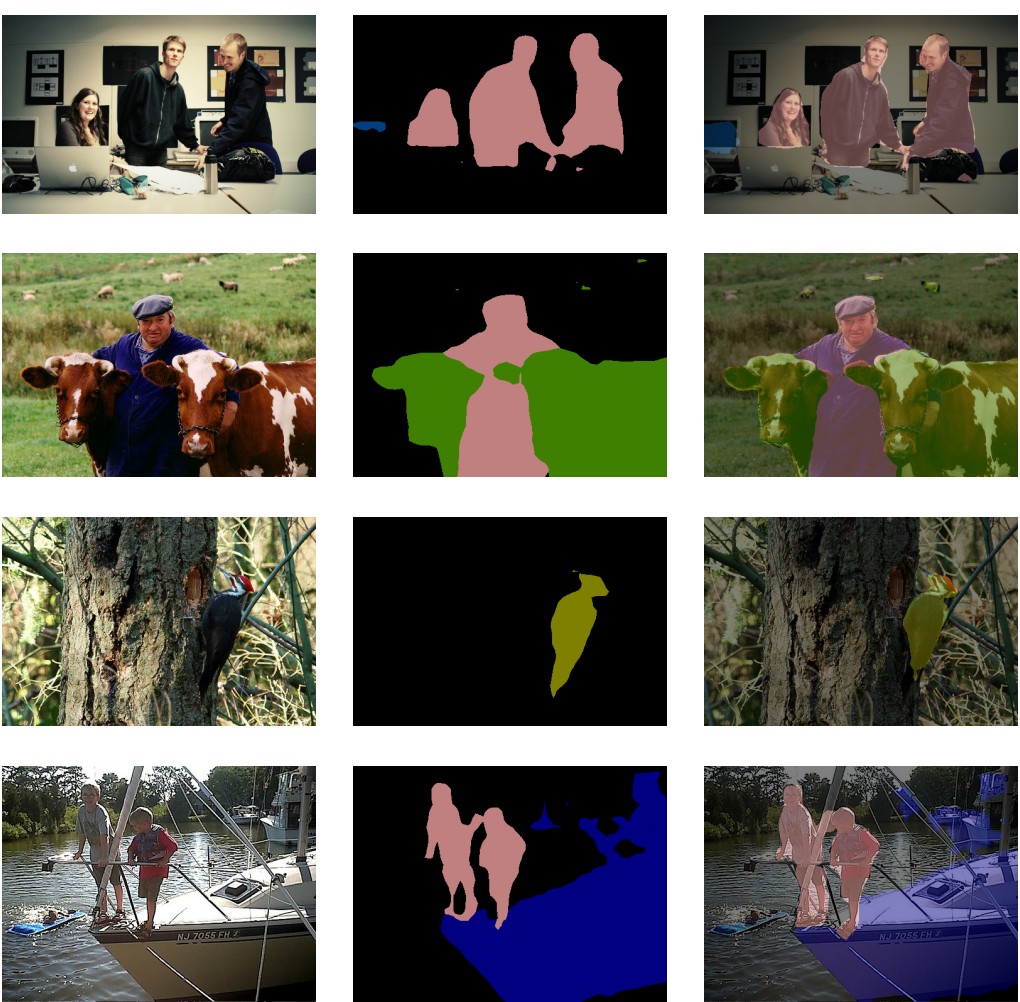

Figure 7: Semantic segmentation results using Deeplabv3 with MobileViTv3-1.0 as its backbone.

