# OpenReview forum: "MobileViTv3: Mobile-Friendly Vision Transformer with Simple and Effective Fusion of Local, Global and Input Features"
_ICLR.cc/2023/Conference — Submitted to ICLR 2023_

### Official Review · Reviewer_AJcM · 2022-10-21

**Confidence:** 5
**Correctness:** 2
**Technical Novelty And Significance:** 2
**Empirical Novelty And Significance:** 3
**Recommendation:** 3

**Clarity, Quality, Novelty And Reproducibility:**

This paper is not well written as there are many unclear details as I point out above. Such issues make it difficult to assess the merit of this work, given that it essentially just proposes a few well-know modifications to existing models without changing the meta architecture at all.

The paper is not well organized. I feel a lot of interesting part should not be put into appendix but in the main body such as latency numbers and limitations. Adding such content into the main body will greatly help improve the quality of the manuscript.

The main text also finishes too abruptly. There is no conclusion or future work, and the ablation study stops suddenly. Please consider better organize the manuscript.

**Strength And Weaknesses:**

It is of great interest to study possible solutions for optimizing transformer models regarding the speed while maintain its superior quality. Although there are abundant works on pushing the boundary of transformer models in vision tasks in terms of quality, there is far less study on optimizing those heavy-lifting models for on-device use cases, where convents such as MobileNet and EfficientNet still dominate.

This work builds on top of two recent works, MobileViT V1 and V2, and proposes simple way to improve the model performance without altering the model architectures significantly. The figures in the paper provide a clear view of the landscape of recent on-device ViT models, and the place of the proposed MobileViT V3 models. This gives users a good sense of how the proposed model compares to existing works. This is highly appreciated.

Technically, this work is sound: using depth wise conv for speed-up and 1x1 conv to shrink model parameters are both widely used techniques to optimize a model. From experiments, the ablation study does show that these modifications indeed contribute to performance improvement.

Despite of the strength, there are a lot of problems in the current manuscript that weakens this work.

1. The paper claims it “addresses the scaling and simplifies the learning task” However, from the experimental results, it is not clear why the proposed modifications can achieve the claimed goal. Even the original MobileViT V1 and V2 are not difficult to scale up (especially V2 as the authors already have several variants of that by scaling it up and down): there are just a matter of changing number of filters or hidden size, and possibly adding or remove certain number of blocks. In addition, I do not see the original MobileViT models are not easy to train. In this sense, this paper’s claim does not hold: it is not supported from the experiments.

2. Another big issue is benchmarking. This paper presents the plots of accuracy v.s. model FLOPs and number of parameters, where the proposed model has fewer number of parameters or FLOPs compared to its rivals and existing convnets. It gives a false impression that the proposed models outperforms all existing convents and ViT models. This is not exactly true. For on-device scenarios, FLOPs and number of parameters are not the golden rule to decide a model’s real performance, but latency is. Especially, although MobileNet has larger number of parameters, it actually runs much faster than ViT model (MobileViT) due to CPU-friendly ops. As pointed in [1] and [2], number of parameters and FLOPs are not positively correlated to latency. In fact, when a model contains excessive transpose/reshape ops or many branches, it can lead to higher latency even though the number of parameters are less.
Unfortunately, the only real latency benchmark is present in Appendix B Table 8, on MobileViT model variants only. It does not provide a holistic picture on the real latency comparison of existing commonly used mobile models.
Therefore, a latency benchmark on real mobile devices (not only on desktop GPU) would be more convincing, just like what MobileViT V1 and V2 have done. Without such benchmark, this work is less convincing that it will run fast as expected.

3. Technical details
There are many unclear details that need further clarification. For example:

    - When replacing 3x3 conv by 1x1 conv, you will have smaller receptive field as 1x1 conv generally serves a channel reduction method. Due to such loss, how do you ensure the information from global features is proper preserved?

    -  How do you decide the scaling factor in Table 1? They seem pretty ad-hoc. For example, 1.3, 1.6 and 1.7 are not quite standard numbers to scale a model.

   -  In implementation details, it is not clear why the model can only be trained with a much smaller batch size. Given that the model has fewer FLOPs and number of parameters comparing to MobileViT V1, it should be trainable using the same batch size as MobileViT V1, which is 1024.

4. Related works miss a few recent works, for example:

    - LeViT: a Vision Transformer in ConvNet's Clothing for Faster Inference, 2021
    - EfficientFormer: Vision Transformers at MobileNet Speed, 2022
    - Next-ViT: Next Generation Vision Transformer for Efficient Deployment in Realistic Industrial Scenarios, 2022

Please consider adding these papers and do a more comprehensive comparison. All released code so it should not be difficult to reproduce those works. This is also related to experimental validation. The current comparison is mostly on MobileViT variants and old convnets. It is far from enough to justify the proposed model superiority. More recent models should be included and compared.

- [1] An Improved One millisecond Mobile Backbone, 2022
- [2] EfficientFormer: Vision Transformers at MobileNet Speed, 2022

**Summary Of The Paper:**

This work proposes a series of modifications to the MobileViT V1 and V2 architectures. It introduces a few tweaks to the original model architecture, including replacing regular conv by depthwise conv to reduce latency, using 1x1 conv instead of 3x3 conv to reduce number of parameters, and adding residual connection in the proposed MobileViT V3 block. Such modifications brings in additional quality gain compared to original MobileViT V1 and V2, while being more efficient in terms of FLOPs and number of parameters in some cases. Experiments on downstreaming tasks such as detection and semantic segmentation also show that the proposed modifications can benefit dense prediction tasks as well.

**Summary Of The Review:**

In summary, this work tries to study a good problem and propose valid modifications to improve existing models. However, it falls short regarding technical details, latency benchmark and comparison with recent works.

I would like authors to address my comments, especially regarding lack of latency benchmark and comparison with more recent models.

---

> ### Author Response · Authors · 2022-11-16
> **Response to reviewer AJcM**
>
> Thank you for the valuable feedback.
>
> Following addresses the concerns:
>
> 1. Regarding Scaling :  In section 3.1, the second motivation for replacing 3x3 convolutional layer with 1x1 convolutional layer, explains how it addresses the scaling challenges: “Scaling MobileViTv1 from XXS to S is done by changing width of the network and keeping depth constant. Changing width (number of input and output channels) of MobileViTv1 block causes large increase in number of parameters and FLOPs. For example, if the input and output channels are doubled (2x) in MobileViTv1 block, the number of input channels to 3x3 convolutional layer inside fusion block increases by 4x and output channels by 2x, because input to the 3x3 convolutional layer is concatenation of input and global representation block features. This causes a large increase in parameters and FLOPs of MobileViTv1 block. Using 1x1 convolutional layer avoids this large increase in parameters and FLOPs while scaling. Table 1 shows that MobileViTv3-S is able to double (2x) the number of channels in most of the layers without drastic increase in number of parameters and FLOPs.”.
> Regarding simpler optimization : We are not claiming that original MobileViT models are difficult to train. Our proposed changes allows MobileViTv3 models to converge faster as seen in table 5.
>
> 2. This work addresses the issue with most of the mobile-friendly models which is drop in accuracy numbers. Our main motivation is not to improve the latency of these small models, but to improve the accuracy while maintaining similar parameters and FLOPs.
>
> 3.
>
> 3.1. When replacing 3x3 conv layer by 1x1, we asked ourselves following fundamental questions: why do we need a 3x3 convolutional layer when fusing local and global information in the fusion block? The goal is to fuse local and global information of each location, then why fuse additional local and global information from other locations in the receptive field using 3x3 conv? This led to the first motivation explained in section 3.1.
>
> 3.2. These numbers were chosen to match the number of parameters of our proposed models to the baselines for accuracy comparisons. Example: Scaling factor of 1.3 and 1.6 were chosen for MobileViTv3-XXS to match parameters with MobileViTv1-XXS. MobileViTv3-XXS has 1.25 million parameters and MobileViTv1-XXS has 1.3 million parameters.
>
> 3.3. As mentioned in section 4.1, due to resource limitations we were only able to train the model on batch size of 384 for MobileViTv3-S and XS. To maintain consistency in batch sizes, MobileViTv3-XXS is also trained on batch size of 384. Table 2 shows that increasing batch size helps to increase accuracy of our proposed models. Therefore, training our proposed models on batch size of 1024 should give further accuracy gains.
>
> 4. Thank you for highlighting these papers. We tried to add all the models to the best of our knowledge in this work. Parameters of LeViT family of models ranges from 7.8M to 39.1M parameters, EfficientFormer from 12.3M to 82.1M parameters, and Next-ViT from 31.7M to 57.8M parameters. Our study focuses on models with 6M or less parameters.
>
> The conclusion/discussion section is present in Appendix D. Due to space limitations we were not able to incorporate it in the main paper.

---

### Official Review · Reviewer_kTXc · 2022-10-25

**Confidence:** 3
**Correctness:** 3
**Technical Novelty And Significance:** 2
**Empirical Novelty And Significance:** 2
**Recommendation:** 3

**Clarity, Quality, Novelty And Reproducibility:**

The motivation of this paper is clear. The work is based on previous publications with a few modifications. Thus, it cloud be easy to reproduce the experimental results.

**Details Of Ethics Concerns:**

There are no ethics concerns in this paper.

**Strength And Weaknesses:**

### Pros:

- The motivation is clear. It’s practical to design a model that is lightweight and efficient on mobile devices.
- The technical comparisons between the MobileViT family are clear, as shown in Figure 2. The intuitive and straightforward illustration makes the methodology easy to follow.
- Extensive experiments (including image/video classifications, object detection, and semantic segmentation) have been provided to validate the effectiveness. However, I still hold some concerns about the experimental comparisons, which are shown in the cons part.

### Cons:

- ***Lack of in-depth analysis and limited technical contribution***

The primary concern is the technical contribution. The components proposed in this paper are the common techniques to achieve the corresponding purpose. For example, adopting 1x1 conv to minimize the network parameters and employing depthwise 3x3 conv to reduce parameters and FLOPs without significantly impacting performance have already been investigated in previous works [1,2,3]. Besides, this paper needs to include an in-depth analysis of the usage of the above modules. From my perspective, I admire the authors’ efforts in the engineering exploration. However, the main ideas have been explored in many previous works, which makes the works seem to be incremental.

- ***Over-claimed contribution and unclear writing***

The paper emphasizes that this work addresses the challenges of scaling and simplifies the learning tasks. However, I can not see it’s an essential problem in the previous MobileViT v1 and v2, since there are also a series of scaled models in MobielViT v1 and v2. The contribution seems to be over-claimed. Please present this part more clearly.

- ***Insufficient comparisons***

The experimental comparisons are mainly conducted on the Top1 accuracy, parameters, and Flops. However, for mobile applications, the actual latency also matters. Please report the throughput for comprehensive comparisons.

- ***Others***

(a) *Extended comparisons:* Since depthwise 3x3 conv is utilized in the MobileViT v3, I would like to see the ablation studies of this module on object detection and segmentation since depthwise conv usually performs well in the dense prediction tasks.

(b) Please add the conclusion part to make the paper more complete.

---

[1] Pyramid Vision Transformer: A Versatile Backbone for Dense Prediction without Convolutions, 2021.

[2] PVT v2: Improved Baselines with Pyramid Vision Transformer, 2021.

[3] Co-Scale Conv-Attentional Image Transformers, 2021.

**Summary Of The Paper:**

This paper introduces a light-weight model for mobile vision tasks, namely MobileViT-v3. The proposed model is based on the previous MobileVit v1 and v2. Specifically, several modifications have been conducted to improve the previous architectures, including replacing 3x3 conv with 1x1 conv, fusing local and global features, adding a residual connection between the input feature and the fusion feature, and replacing normal 3x3 conv with depthwise 3x3 conv. The above changes result in more efficient and effective mobile backbones. Experiments have been performed on the MobileViT families to validate the effectiveness.

**Summary Of The Review:**

Overall, the paper provides extensive experiments to verify its effectiveness. The motivation is clear. I admire the authors’ efforts in engineering exploration. However, the technical novelty is limited since the previous works have investigated the core components (i.e., 1x1 conv, depthwise 3x3 conv, and residual connection). Besides, the paper lacks in-depth analysis. I will give my first rating as `reject`.

---

> ### Author Response · Authors · 2022-11-16
> **Response to reviewer kTXc**
>
> Thank you for the valuable feedback.
> Following addresses the concerns:
>
> 1. As explained in section 3.1, the two main motivations of using 1x1 conv are not to reduce the parameters, but to 1. fuse local and global features independent of other locations in the feature map to simplify the fusion block’s learning task and 2. To remove one of the major constraints in scaling of MobileViTv1 architecture. We are not proposing ways to reduce parameters, but a new module capable of learning better features. Reduction of parameters is an inherent advantage of our approach.
>
> 2. In section 3.1, the second motivation for replacing 3x3 convolutional layer with 1x1 convolutional layer, explains how it addresses the scaling challenges. Scaling MobileViTv1 from XXS to S is done by changing width of the network and keeping depth constant. Changing width (number of input and output channels) of MobileViTv1 block causes large increase in number of parameters and FLOPs. For example, if the input and output channels are doubled (2x) in MobileViTv1 block, the number of input channels to 3x3 convolutional layer inside fusion block increases by 4x and output channels by 2x, because input to the 3x3 convolutional layer is concatenation of input and global representation block features. This causes a large increase in parameters and FLOPs of MobileViTv1 block. Using 1x1 convolutional layer avoids this large increase in parameters and FLOPs while scaling.
>
> 3. We have provided latency comparison between MobileViTv1 and MobileViTv3 family of models in table 8.
>
> 4. The conclusion/discussion section is present in Appendix D. Due to space limitations we were not able to incorporate it in the main paper.

---

### Official Review · Reviewer_8bQB · 2022-10-25

**Confidence:** 5
**Clarity, Quality, Novelty And Reproducibility:** 1) easy to read and follow, and easy …
**Correctness:** 3
**Technical Novelty And Significance:** 2
**Empirical Novelty And Significance:** 2
**Recommendation:** 5

**Strength And Weaknesses:**

Strength:

1) From fig3, Performance of mobilevitv3 is significantly improved comparing with v2/v1, when # of  parameters is smaller than 3.5m.

2) The model is well evaluated based on various tasks:  imagenet for classification, coco for detection and pascal ade20k for segmentation. In all tasks the model shows significant improvement especially on XS cases.

3) Extensive related works are studied in the paper such that the results are compared against strong enough SoTA methods.

Cons:

The ablation study of modifications is lacked in the paper, i.e. step-by-step performance [flops, parameters, acc] comparison of doing each modification, so that we may got the idea of the contribution of each part.

The novelty of the paper is somehow limited since most modification is already proposed in prior works [depth-wise conv etc.], and the improvement is marginal when paraemters > 3.5m on imagenet.



**Summary Of The Paper:**

The paper propsed mobilevit v3 which did modification based on mobilevit by reducing conv3x3 to conv1x1, adjusting the routing from input and pre-conv embedding [local feature as named in the paper] layers for feature fusion.

It produces a better efficiency especially at low-parameter cases.

**Summary Of The Review:**

good performance against SoTA baselines, while the modification and pruning of network follows prior arts with limited novelty.

---

> ### Author Response · Authors · 2022-11-16
> **Response to reviewer 8bQB**
>
> Thank you for the valuable feedback.
>
> In Table 5 we provide ablation study for accuracy and in table 6 for FLOPs and parameters.

---

### Official Review · Reviewer_jAum · 2022-11-01

**Confidence:** 5
**Correctness:** 4
**Technical Novelty And Significance:** 1
**Empirical Novelty And Significance:** 2
**Recommendation:** 3

**Clarity, Quality, Novelty And Reproducibility:**

The novelty seems to be very limited. It builds upon previous MobileViT architectures with little to no major contribution. The proposed effort seems to be reproducible as authors have mentioned about public release of the code in GitHub.

**Strength And Weaknesses:**

Strengths:

1. The paper is well-written and easy to follow. The visualizations allow for better understanding of the proposed technique.

2. The proposed method has been rigorously tested on many different tasks and datasets.

3. The authors have performed many optimizations to further perfect the proposed blocks. This is reflected in consistent improvements as well as the reduction in number of parameters and flops

Weaknesses/Concerns

1. My main criticism of this work is lack of novelty. As mentioned above, this work is an amazing engineering effort through which various building blocks (e.g. 3 x 3 conv layers, etc.) have been replaced with seemingly better alternatives. However, it is hard to pin-point a major contribution or novelty which is specific to this work (e.g. novel design of an attention layer)

2. It is understandable that authors attempt to further optimize the already existing MobileViTv1 and MobileViTv2 blocks. However, one may wonder why the transformer layers should  proceed the conv-based layers ? why not processing features in parallel streams ? the current design is very limited in novelty, and the motivation behind it seems to be the reduction of number of flops, parameters, etc. However, there should also be better justification of the intuition.

3. Experimental comparisons are presented in a somewhat contrived settings. It is preferred to know how the proposed method performs compared to other competing approaches.

4. Performance gains in downstream tasks of semantic segmentation on ADE20K dataset and object detection on COCO are higher than classification. What could be the reason behind this ?

5. Another concern is the throughput of the model with the proposed blocks on devices other than GPU. Is there any comparison on mobile devices ? in fact, the real latency on such devices could be drastically different than what is measured on the GPU. In addition, it is recommended to use more modern GPU hardware (e.g. A100) to report latency numbers to better illustrate the performance differences.

**Summary Of The Paper:**

This paper attempts to resolve the scaling issues of MobileViTv1-blocks which hinder the learning. Specifically, the authors propose MobileViTv3-block which has a more simplified architecture. The proposed blocks have been added to both MobileViTv1 (with transformer blocks) and MobileViTv2 (with linear transformer blocks) models and experiments on various tasks such as classification (ImageNet), detection(COCO) and segmentation (ADE20K) demonstrate improved results.

**Summary Of The Review:**

This work presents new MobileViT blocks that make the architecture more efficient and purportedly more accurate in different tasks (e.g. classification, segmentation, detection, etc.). Despite the great engineering effort behind this work, the novelty and contribution are very limited and as a results don't merit acceptance.

---

> ### Author Response · Authors · 2022-11-16
> **Response to reviewer jAum**
>
> It is true that we started from MobileViT-block, but after our proposed architectural changes, we have engineered a new ViT based module. Also it is clear from the paper that our module is able to achieve higher accuracies with similar parameters and FLOPs. To the best of our knowledge, we have compared our models with existing state of the art light-weight models (both CNN and ViT based). Moreover, we extended our work from classification to segmentation and detection and showed the effectiveness of our module. The architectural changes, especially the choice of 1x1 conv instead of 3x3 conv to combine local and global information helps optimization of our model. As a result of this, the rich features learnt not only allow better classification, but also lead to better performance of downstream tasks. Due to hardware (GPU) limitations, the latency numbers were generated on available GPUs.

---

### Author Response · Authors · 2022-11-16
**Updated paper**

Unit correction in Table 2, 6, and 7:  Model FLOPs unit was erroneously mentioned as ‘G’ (10e9). The correct unit for FLOPs is ‘M’ (10e6) and has been updated in the tables.

---

### Decision · Program_Chairs · 2023-01-20

**Decision:**

Reject

**Justification For Why Not Higher Score:**

The limited novelty makes the current work not reach the bar.

**Justification For Why Not Lower Score:**

N/A

**Metareview: Summary, Strengths And Weaknesses:**

This submission receives all negative reviews. The raised issues include a lack of novelty, unclear technical presentations, and insufficient experimental validations. Although the authors try to address these issues during the rebuttal phase, the novelty issue still remains a crucial point to limit this work from being recognized. After checking all the reviews and rebuttals, the AC agrees to the reviewer that the current form is not ready for publication. The authors are suggested to further improve the current work and welcome to the next venue.